# GL-Fusion: Rethinking the Combination of Graph Neural Network and Large Language Model

## Abstract

Recent research on integrating Large Language Models (LLMs) with Graph Neural Networks (GNNs) typically follows two approaches: LLM-centered models, which convert graph data into tokens for LLM processing, and GNN-centered models, which use LLMs to encode text features into node and edge representations for GNN input. LLM-centered models often struggle to capture graph structures effectively, while GNN-centered models compress variable-length textual data into fixed-size vectors, limiting their ability to understand complex semantics. Additionally, GNN-centered approaches require converting tasks into a uniform, manually-designed format, restricting them to classification tasks and preventing language output. To address these limitations, we introduce a new architecture that deeply integrates GNN with LLM, featuring three key innovations: (1) Structure-Aware Transformers, which incorporate GNN's message-passing capabilities directly into LLM's transformer layers, allowing simultaneous processing of textual and structural information and generating outputs from both GNN and LLM; (2) Graph-Text Cross-Attention, which processes full, uncompressed text from graph nodes and edges, ensuring complete semantic integration; and (3) GNN-LLM Twin Predictor, enabling LLM's flexible autoregressive generation alongside GNN's scalable one-pass prediction. GL-Fusion achieves outstand performance on various tasks. Notably, it achieves state-of-the-art performance on OGBN-Arxiv and OGBG-Code2,

## 1 Introduction

Research in Graph Neural Networks (GNNs) has long focused on learning from graph with pre-processed vector features, often overlooking the rich textual information contained in raw data. Recently, many studies have recognized that better utilization of these text features can enhance performance. This has led to a focus on text-attributed graphs (TAGs), graphs with node, edge, and graph-level text attributes. In addition, some tasks may contain a task description, questions, and candidate answers in natural language. The combination of GNNs and pretrained Large Language Models (LLMs), which can efficiently encode these text attributes and information, has garnered significant interest.

Two main approaches have emerged for combining GNNs and LLMs: GNN-centered methods and LLM-centered methods. GNN-centered methods focus on typical graph tasks like node classification and link prediction. These methods convert text into representation vectors for graph nodes or edges using LLMs, which are then fed into GNNs to produce final predictions. GNN-centered approaches excel at capturing structural information. However, compressing rich textual features into fixed-length vectors results in information loss. Moreover, The importance of different text features varies depending on the task. For example, in wiki knowledge graph (KG) datasets, entity descriptions may include a person's name, nationality, or occupation. A name might be crucial for identifying relatives but irrelevant for finding their workplace. Additionally, GNN architectures cannot generate natural language, making it difficult to perform tasks like question answering.

LLM-centered methods, on the other hand, use a multi-modal approach by projecting graph representations into the language space, allowing LLMs to generate text answers to graph-text questions.

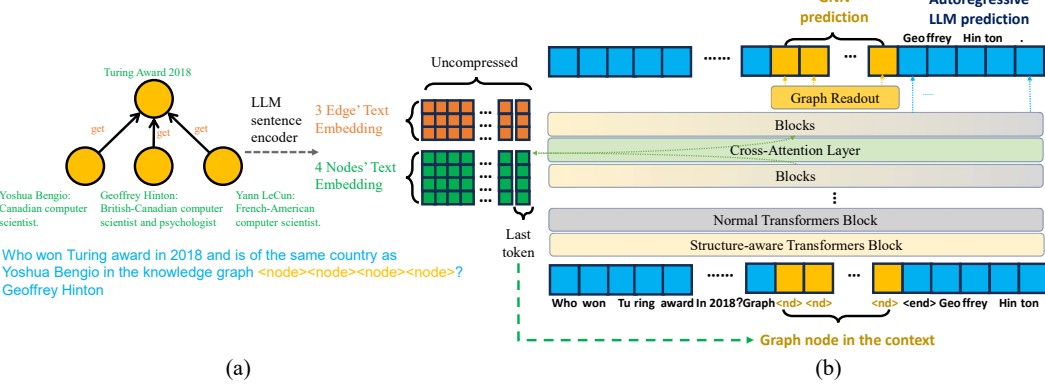

Figure 1: Workflow of the GL-Fusion model. (a) An example of a text-attributed graph, where each node and edge has a text attribute that will be encoded by the LLM. (b) The encoded text-attributed graph nodes (<node> in the figure), along with a prompt or question in natural language, are merged into an input sequence for our GL-Fusion model. GL-Fusion consists of several structure-aware transformer blocks and standard transformer blocks. Cross-attention layers are inserted to retrieve the complete, uncompressed node and edge text. The output on graph nodes is further processed by graph readout components, while the output on standard text tokens predicts the next token, similar to an original autoregressive LLM.

However, this approach faces similar challenges. Graph tasks vary widely in domains, and current LLM-centered models lack dynamic encoding by GNNs that adapt to tasks, so these models struggles to generalize to diverse datasets or tasks. Furthermore, language supervision is often indirect and less efficient compared to direct graph supervision, making training more difficult. For example, LLMs, with their autoregressive nature, cannot simultaneously predict all nodes in a graph, unlike GNNs, which can process nodes in parallel. Moreover, GNNs focus on classifying nodes into a finite set of classes, while LLMs operate in the much larger and noisier token space of natural language, complicating the training process.

To handle tasks involving both graphs and text, we introduce Graph-Language Fusion (GL-Fusion), an architecture that combines the strengths of GNN-centered and LLM-centered models. As shown in Figure 1(a), graphs are paired with node and edge text features to solve tasks described in natural language. In Figure 1(b), the task description is included as part of the input text sequence, and a special token <node> is added for each graph node. Each node's feature is generated by a text encoder based on its text attribute, which is then incorporated into the sequence. To process this combination of graph and text tokens, our architecture includes:

- **Structure-Aware Transformer Layers**: We modify the original attention layers for language input to graph structure-aware attention with both graph and text inputs, which maintains causality for language generation and further enabling transformer to encode graph structure. Different from graph transformer, which encodes graph only and cannot process text, our transformer can naturally encodes graph tokens conditioning on the text tokens before it in sequence, and encodes text tokens after the graph based on the graph. This two-way information exchange between graph and text overcomes the limitations of older models, where either graph or text embedding is irrelevant from the other, and achieves better expressivity.

- **Graph-Text Cross-Attention**: Previous models often compress either graph or text information into a single vector, leading to information loss. In our model, graph structure is preserved across each transformer layer without compression. To avoid compressing textual features, we introduce Graph-Text Cross-Attention layers, where attention is applied between node text and the main sequence (which includes the <node> tokens). These layers retain the complete text information of each node, allowing the model to more precisely extract task-relevant information by focusing on the most important tokens.

- **GNN & LLM Twin-Predictor**: Different tasks require different prediction mechanisms. For traditional GNN tasks, such as node classification, having the LLM generate predictions for each node would slow down inference and be constrained by graph size. In such cases, direct readout from the GNN is more efficient. Conversely, tasks that require natural language outputs benefit from the autoregressive capabilities of LLMs. Unlike previous models, which rely on either a GNN or LLM predictor, GL-Fusion retains both as predictors. During training, supervision can be applied to both models, and during inference, both outputs are generated simultaneously. Users can choose the most appropriate output for the task or consider both outputs together.

With these modifications, we propose GL-Fusion, a unified architecture that achieves competitive performance by integrating structural and textual information. To demonstrate this, we conducted experiments on various tasks, including basic graph property prediction, node classification, knowledge graph completion, commonsense question answering, and code graph-to-text generation. Our GL-Fusion achieves outstanding performance across all these datasets, outperforming various combinations of GNNs and LLMs. Notably, our model even achieves state-of-the-art performance on two Open Graph Benchmark (Hu et al., 2020) datasets: ogbn-arxiv and ogbg-code2. These results demonstrate the powerful capability of our GL-Fusion architecture.

## 2 PRELIMINARY

For a tensor $Z \in \mathbb{R}^{a \times b}$, let $Z_i \in \mathbb{R}^b$ denote the $i$-th row, $Z_{:,j} \in \mathbb{R}^a$ denote the $j$-th column, and $Z_{ij} \in \mathbb{R}$ denote the element at the $(i, j)$-th position.

**Text-Attributed Graph (TAG)** A TAG is represented as $\mathcal{G} = (V, E, X)$, where $V = 1, 2, 3, \ldots, n$ is the set of $n$ nodes, $E \subseteq V \times V \times \mathcal{Z}^{L_e}$ is the set of edges. Each edge $(i, j, E_{ij}) \in E$ connects node $i$ and node $j$ with text $E_{ij}$ consisting of $L_e$ tokens. The node text feature matrix is denoted as $X \in \mathbb{Z}^{n \times L_n}$, where $X_i$ represents the $L_n$-token text feature of node $i$. Here, $L_n$ and $L_e$ represent the maximum lengths of node and edge text, respectively. Text of varying lengths is padded to the same length.

**Problem Settings** We consider a TAG $\mathcal{G}$ and a task description $T \in \mathcal{Z}^{L_t}$. The task can be node classification, link prediction, graph classification, or text generation. As illustrated in Figure 1, our model processes a TAG along with a task description. The graph is integrated into the task description as part of the input sequence to our model, where the graph forms a subsequence starting with a special <graph_start> token and ending with a <graph_end> token. Each node is represented by a <node> token. For each specific task, we use a task-specific prediction head, and the model also leverages the LLM itself to generate the answer in text form. We focus on the supervised learning setting, where the dataset is split into training, validation, and test sets. Models are trained on the training set, hyperparameters are tuned on the validation set, and test set performance is reported.

## 3 GL-FUSION: A HIGHLY INTEGRATED GNN-LLM MODEL

This section presents the architecture of our GL-Fusion model, which integrates structure-aware transformer layers, graph-text cross-attention blocks, and GNN & LLM twin predictors. Given input sequence $T \in \mathbb{Z}^{L_t}$ and input graph $\mathcal{G} = (V, E, X)$, we first convert $T$ to sequence representation $t \in \mathbb{R}^{L_t \times d}$ with token embedding layer. Then we use an existing text encoder (BehnamGhader et al., 2024) to convert node text sequences $X$ to uncompressed text embeddings $x \in \mathbb{R}^{n \times L_n \times d}$, and we also convert edge text $E_{ij} \in \mathbb{Z}^{L_e}$ to compressed text embedding $e_{ij} \in \mathbb{R}^d$. Our GL-Fusion model will update input sequence representations $t$ in each transformer layer and use $t$ to produce the final prediction.

### 3.1 STRUCTURE-AWARE TRANSFORMER LAYER

To encode the input sequence, a mixture of text and node tokens, we propose a structure-aware transformer layer that fulfills the following properties: (1) Preserving causality for text generation. (2) Maintaining invariance to node permutation. (3) Encoding edges between nodes. Figure 2a illustrates this structure-aware architecture. Compared to ordinary transformer layers, it includes the following extensions.

**Graph-aware attention mask and positional encodings** To preserve causality for text generation, a causal mask keeps only the lower triangle of the attention matrix, setting all other entries to zero. In other words, each token can only aggregate embeddings from previous tokens, prohibiting the use of representations from tokens that come after it. While directly applying the causal mask to the mixed sequence preserves causality for text generation, it disrupts permutation equivariance; that is, each graph token can only aggregate information from graph tokens preceding it, making the

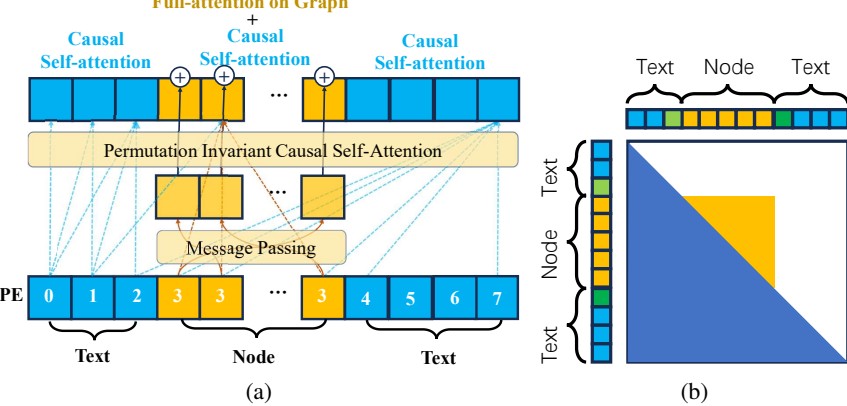

Figure 2: Design of Structure-Aware Transformer layer. (a) Structure-aware Transformer layer. The brown curve indicates the message-passing process along graph edges. The dashed brown and blue lines represent causal self-attention and full attention, respectively. The boxed numbers indicate shared positional encodings. (b) The attention mask in structure-aware transformer layers. The blue part represents the ordinary causal attention mask, and the yellow part allows attention between nodes in the same graph.

order of graph tokens affect the output. This violates the permutation invariance of node indices, which is crucial for graph learning.

To address this, we propose the following attention masks. All tokens can aggregate information from any token before them. However, for node tokens, other node tokens within the same graph are also visible to them, preserving permutation invariance. Additionally, causality for text generation is maintained as graph tokens are not generated. Furthermore, to preserve causality, we only allow graph tokens to attend to text tokens that come before them, and text tokens can only attend to graph tokens that come before them. The resulting attention mask is shown in Figure 2b.

To further ensure permutation invariance, we assign the same positional encodings to all node tokens in the same graph as the corresponding <graph_start> token. Additionally, all graph tokens use a single index, as shown in the positional encoding (PE) of Figure 2a, preventing the model from running out of the context window for large graphs.

**Message Passing with Multiple Aggregators.** To encode graph structure, we involve a message passing layer (Gilmer et al., 2017) in our transformer layer. Our message-passing layer updates the representation of node $u$ as follows:

$$h'_u = \text{COMBINE}(h_u, \text{AGGREGATE}(\{h_v \odot e_{uv} | v \in N(u))), \tag{1}$$

where $h_u \in \mathbb{R}^d$ is the representation of node $u$, produced by a linear layer with the corresponding node token representation $t_i$, $e_{uv} \in \mathbb{R}^d$ is the edge feature extracted from text, and $N(u)$ denotes neighboring nodes. Following Corso et al. (2020), we employ three aggregators: mean, max, and std, concatenating their outputs into a $\mathbb{R}^{3d}$ vector, which is then projected back to $h'_u \in \mathbb{R}^d$ via a linear layer.

To manage the varying importance of graph structure information across tasks, we introduce a gating mechanism. The gate value is computed with a linear layer and a tanh activation function with the original token embedding as input, updating $t_i$, the representation of node token $i$, corresponding to node $u$ as:

$$t_i \leftarrow t_i + \tanh(W t_i) h_u, \tag{2}$$

where $W \in \mathbb{R}^{1 \times d}$. All elements in $W$ are initialized as 0 and optimized in the training process. So the gate value and thus the output of Message Passing module is 0 initially. This mechanism allows gradual integration of GNN outputs during training while stabilizing the model and mitigating knowledge forgetting.

### 3.2 GRAPH-TEXT CROSS-ATTENTION

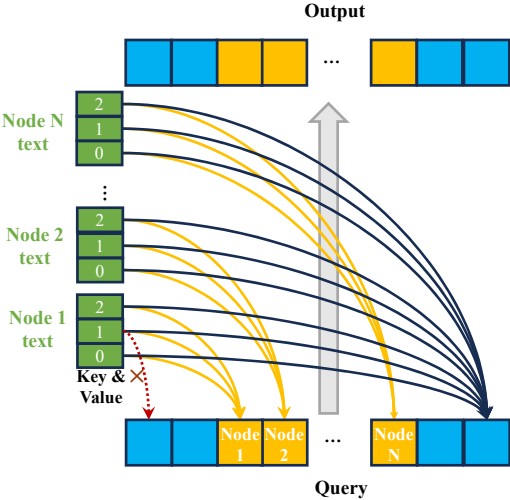

One significant drawback of previous GNN-centered models is that they compress node, edge text features, and task descriptions into fixed-size representations, while LLM-centered methods tend to compress graph structure into fixed-size representations, leading to significant information loss. With our structure-aware transformer layer, we encode the graph structure and task description directly in each layer without compression. To avoid compression of node/edge text features, we introduce the Graph-Text Cross-Attention block to enable node token representations and text tokens generated after the node token to extract information from the raw node text. Note that we can also use the cross-attention block to extract information from edge text. However, most datasets currently have simple and unvaried edge features, so we primarily focus on extracting information from raw node text.

Figure 3: The attention mask in cross-attention layers. For text tokens before the graph, they do not involve cross-attention to maintain causality (red dashed line with the ×). For node tokens `<node>`, each token only has access to its own text (orange lines). For text tokens after the graph, they have access to all node text (black lines).

The architecture is shown in Figure 3. The cross-attention block extracts features from the node text representation $x$ to update the task representation $t$ as follows.

To update $t_i$, the representation of token $i$, first, each node's text representation of $L_n$ tokens is aggregated into a single representation. For node $v$,

$$x'_{iv} = \text{softmax}\left(\frac{1}{\sqrt{d}} t_i^T W_{Q_1}^T W_{K_1} x_v^T\right) x_v \in \mathbb{R}^d, \tag{3}$$

where $t_i \in \mathbb{R}^d$ is the query of cross attention, $W_{Q_1}, W_{K_1} \in \mathbb{R}^{d \times d}$ are learnable linear layer for query and key, and $x_v \in \mathbb{R}^{L_n \times d}$ is the uncompressed text representation of node $v$. $x'_{iv}$ is the feature extract from node $v$ for updating token $i$'s representation. Then, all nodes' features are further aggregated:

$$t'_i = \begin{cases} x'_{iv} & \text{if } i \text{ is the node token w.r.t. node } v, \\ \text{softmax}(t_i^T W_{Q_2}^T W_{K_2} x'_{i\mathbf{v}}) x'_{i\mathbf{v}} & \text{if } i \text{ is a text token,} \end{cases} \tag{4}$$

where $\mathbf{v}$ is the set of nodes whose token in input sequence is before $i$, $x'_{i\mathbf{v}} \in \mathbb{R}^{|\mathbf{v}| \times d}$ is the sequence of $x'_{iv}$ for node $v$ in $\mathbf{v}$, and $W_{Q_2}, W_{K_2} \in \mathbb{R}^{d \times d}$ are learnable linear layers. Then $t'_i \in \mathbb{R}^d$ is used to update $t_i$. To maintain causality, we only allow text tokens to aggregate features from graph nodes preceding them in the input sequence. For text tokens, if there is no graph token before, the resulting cross-attention result is 0. For node tokens, although they can aggregate text features from other nodes, in experiments we find that this may cause all nodes to have similar representations, which makes them indistinguishable from each other. Therefore, we constrain each node to extract information solely from its own text to avoid this problem.

Compared to the cross-attention block, directly adding node text to the input sequence of the LLM may seem more straightforward. However, it introduces several challenges. First, it is difficult to design a special positional encoding (PE) and attention mask that preserve causality for text and permutation invariance for graphs in the structure-aware layer. Second, in terms of computational complexity, the naive text-in-context method incurs $O((nL_n + L_t)^2)$ time complexity as $n$ $L_n$-length node text sequences are added to the context window. In contrast, our cross-attention method simplifies the complexity to $O(nL_n \cdot L_t)$, significantly reducing costs and enabling our model to handle larger graphs efficiently, as is often the case when $nL_n >> L_t$.

### 3.3 GNN & LLM Twin Predictor

Since our model uses a transformer architecture and maintains causality, it can naturally generate text outputs, which we refer to as the LLM predictor. Additionally, since graph nodes are treated

as tokens in the input sequence, the transformer's output representations for these tokens can be considered node-level GNN representations, which can be fed into pooling layers and linear layers to produce predictions—referred to as the GNN predictor. Both predictors have their advantages and disadvantages, as outlined below:

**Rationale for GNN as a Predictor.** While LLMs can perform various language tasks, they face limitations in autoregressive generation, including:

- Natural Language Output: LLMs can natively produce text predictions, which is challenging for the GNN predictor.
- Numerical Output: GNN predictors naturally output numerical values, making them well-suited for regression or ranking tasks. In contrast, LLMs produce token sequences, making it difficult to generate numerical outputs as text.
- Scalability: LLMs predict outputs one by one in an autoregressive manner, which can be inefficient for tasks requiring multiple predictions for all nodes or edges. In contrast, GNNs generate all predictions for all nodes or edges in parallel.
- Training Efficiency: Training models end-to-end through LLM-generated sequences is more indirect and harder to control, as the LLM's loss function is constrained by "perplexity" (i.e., the cross-entropy loss on token space), which often includes task-irrelevant tokens instead of focusing solely on the task-relevant labeling space. With GNNs, we can provide direct and dense supervision on the target classes, improving training efficiency.

To leverage both approaches, we employ a twin-predictor architecture, allowing for simultaneous predictions from both the GNN and LLM. A graph readout component, tailored to specific tasks, is added after the final layer for GNN predictions. This component is trained using cross-entropy loss for classification tasks or mean squared error for regression tasks. During inference, it can generate predictions for graph nodes as needed.

## 4  RELATED WORK

**LLM-Centered Approaches** LLM-centered methods leverage pretrained language models to handle graph data by translating graph structures into text formats, like adjacency lists (Liu & Wu, 2023; Guo et al., 2023b; Chen et al., 2024). While these methods have shown some promise, they often suffer from issues related to node and edge ordering and can struggle with large graphs due to the limited context windows of LLMs. A more effective solution has been to use GNNs to generate node representation sequences that are fed into LLMs. For example, Chai et al. (2023) used embeddings from Message Passing Neural Networks (MPNNs) to represent target nodes, enabling the LLM to answer basic structural questions. Similarly, Tang et al. (2024) aligned both structural and textual inputs using GNNs and Transformers, achieving better accuracy by processing all node embeddings through a frozen pretrained LLM. However, these approaches can still face limitations, such as task-agnostic encoders, as seen in Qin et al. (2024), which limit their ability to transfer across different domains.

**GNN-Centered Approaches** In GNN-centered methods, LLMs are primarily used to encode textual data that is then fed into GNNs for tasks like node classification and link prediction. Early graph datasets such as Cora (McCallum et al., 2000) and CiteSeer (Giles et al., 1998) used basic text embeddings, but these often failed to capture complex textual information. More recent datasets, like OGB-WikiKG90Mv2 (Hu et al., 2021), have improved this by utilizing pretrained sequence encoders. Duan et al. (2023) demonstrated that fine-tuning LLMs before passing textual information to GNNs significantly boosts performance. Further work by Ioannidis et al. (2022) and Xie et al. (2023) introduced novel fine-tuning strategies to better integrate LLMs into GNN tasks. Besides, with many powerful LLMs being closed-source, researchers like He et al. (2024) have proposed augmenting graph data with enhanced text features generated by accessible LLMs. While they improve performance, these models like OFA (Liu et al., 2023) still compress textual data into fixed-length vectors, limiting their application to tasks with complex text features.

**GNN-LLM Fusion** Recent efforts aim for a deeper integration of GNNs and LLMs. One example is GraphFormers (Yang et al., 2023), which proposed an architecture that combines GNNs and

Table 1: Results in accuracy (%) on Basic Graph Property Prediction Tasks.

|           | GL-Fusion | EdgePrompt | GML   | GraphML | w/o Cross Atten |
|-----------|-----------|------------|-------|---------|-----------------|
| degree    | **100**   | 44.87      | 20.91 | 40.20   | 100             |
| edge      | **99.78** | 74.60      | 50.45 | 62.05   | 100             |
| node text | **34.50** | -          | -     | -       | 0.00            |

Table 2: Results in accuracy (%) on node classification Tasks.

|           | GL-Fusion | GLEM  | XRT   | OneForAll | GPT4graph | GraphGPT | GCN   |   |
|-----------|-----------|-------|-------|-----------|-----------|----------|-------|---|
| Ogbn-arxiv | **78.20** | 76.12 | 76.94 | 77.51     | 60.00     | 75.11    | 71.47 |   |
| Cora      | **84.3**  | -     | -     | 74.76     | -         | -        | 81.5  | - |

Transformers by iteratively encoding text and aggregating graph structures across layers. While this approach improves text encoding for graph nodes, it overlooks the inclusion of task descriptions or language prompts. Furthermore, the lack of autoregressive generation and cross-attention mechanisms restricts the model's ability to dynamically extract task-specific information. Building on this, Jin et al. (2023) adapted the GraphFormers architecture by refining its pretraining strategy, aiming to better capture text relationships between neighboring nodes. While these methods have made strides, there is still a need for more tightly integrated approaches that can fully exploit the potential of GNNs and LLMs working together.

## 5 EXPERIMENTS

To evaluate the potential of GL-Fusion as a new architecture combining GNN and LLM, we conducted experiments on various tasks. These include synthetic tasks to validate its capacity to capture basic graph properties, traditional GNN tasks such as node classification and link prediction to assess its ability to solve graph-related tasks, commonsense question-answering tasks to test its ability to leverage knowledge graphs for language generation, and code graph tasks to evaluate its capacity to generate text based on graph structures. Through these experiments, we demonstrate GL-Fusion's strong potential to effectively combine GNN and LLM architectures. Further details on the experiments can be found in Appendix A.

### 5.1 BASIC GRAPH PROPERTY PREDICTION

Following Guo et al. (2023a), we evaluate our model on basic graph property prediction tasks. We consider two tasks: degree (to predict a node's degree with the whole graph as input) and edge (to predict whether there exists an edge between two nodes in the graph). These two properties are simple for GNN-centered methods. However, for LLM-centered methods, graph structural features are often compressed and thus incomplete, leading to difficulties. The baselines include LLM-centered methods such as EdgePrompt, GML, and GraphML from Guo et al. (2023a). We also conduct an ablation study on these two tasks. Additionally, we introduce a node text retrieval task, where models attempt to predict the input sentence of a node. The results are shown in Table 1. In general, GL-Fusion outperforms all these baselines and captures basic graph properties perfectly. Moreover, GL-Fusion without cross-attention results in significantly lower precision on the node text task, validating the effectiveness of our cross-attention block in extracting information from node text.

### 5.2 NODE CLASSIFICATION

We demonstrate our model's ability to leverage both textual features and graph structure in TAG through a node classification task. We evaluate GL-Fusion on two datasets: ogbn-arxiv (Hu et al., 2020) and Cora (Yang et al., 2016). The baselines include GCN (Kipf & Welling, 2017), the GNN-centered model GLEM (Zhao et al., 2023), XRT (Chien et al., 2022), OFA (Liu et al., 2023), and the LLM-centered methods GPT4graph (Guo et al., 2023a), MuseGraph (Tan et al., 2024), and GraphGPT (Tang et al., 2024). The results are shown in Table 2. We also conduct experiments in

Table 3: Results in test accuracy(%) on ogbn-arxiv dataset with a few training samples.

| #shots | Feature | PLM+GAE | GIANT | PLM-cls | PLM-dense | PLM-sparse | G-Prompt | GL-Fusion |
|--------|---------|---------|-------|---------|-----------|------------|----------|-----------|
| 10 | 45.76 | 51.89 | 51.4 | 46.97 | 51.17 | 52.01 | 52.48 | **56.44** |
| 100 | 58.75 | 60.63 | 61.26 | 58.69 | 58.65 | 60.85 | 61.67 | **68.18** |

Table 4: Node classification on CSTAG datasets (Yan et al., 2023). Sports uses F1 score. Other datasets use accuracy.

| | | PLM-Based | | GNN-Based | | | | Co-Training Based | |
|-------|-----------|-------|-------|-------|-------|--------|--------|--------|---------|
| Model | GL-Fusion | Tiny | Base | T-GCN | B-GCN | T-SAGE | B-SAGE | GCN(T) | SAGE(T) |
| Children | **61.19** | 49.85 | 59.91 | 57.07 | 58.11 | 57.57 | 58.74 | 54.75 | 59.7 |
| History | **86.16** | 83.06 | 86.09 | 84.52 | 85.04 | 84.79 | 85.12 | 83.52 | 85.09 |
| Photo | 82.89 | 73.75 | 77.53 | 82.42 | 82.7 | 83.25 | 83.27 | 83.32 | **86.64** |
| Computers | **88.91** | 58.32 | 60.4 | 87.43 | 87.86 | 87.9 | 88.3 | 83.93 | 86.04 |
| Sports | **93.33** | 81.47 | 86.02 | 84.93 | 86.16 | 87.06 | 87.34 | 85.06 | 85.87 |

few-shot settings following (Huang et al., 2023). The results are shown in Table 3. #shot means the number of training set per class. Baselines are from Huang et al. (2023). Our model outperforms existing models significantly. Besides these datasets, we also conduct experiments on CSTAG benchmark (Yan et al., 2023). We use the datasets and baseline in (Yan et al., 2023). PLM-Based models uses LLM to encode target node text only. GNN-Based methods uses GNN to encode graph with node feature provided by fixed LLM. Co-Training methods denote methods training GNN and LLM simultaneously. The results are shown in Table 4. Our GL-Fusion significantly outperforms all baselines on most datasets, verifying the model's capacity for ordinary graph tasks.

## 5.3 KNOWLEDGE GRAPH COMPLETION

We demonstrate our model's ability to leverage both textual and structural information through the knowledge graph completion task, using the Wikidata Knowledge Graph, which provides rich textual descriptions and structural relationships. We employ the FB15k-237-ind dataset (Teru et al., 2020), extracted from Freebase (Bollacker et al., 2008), featuring four standard training and test splits with shared relation types but disjoint entities. Each node and edge is annotated with textual attributes: entities include names and brief descriptions, while edges retain their textual representations from Freebase. Following approaches like NBFNet (Zhu et al., 2022) and UniLP (Mikhail et al., 2024), we annotate nodes with distances to their corresponding head or tail nodes for prediction tasks.

For baselines, we select several representative inductive learning GNNs and KG-BERT (Yao et al., 2019) for LLM-based completion. Recent methods using LLMs for KG completion, such as few-shot prompting (BertRL)(Zha et al., 2021) or explicit rule learning (KRST)(Su et al., 2023), are also included as they focus on different techniques. Our results, shown in Table 5, indicate that our model outperforms all baselines across the four splits, effectively utilizing both linguistic and structural information.

## 5.4 COMMON SENSE QUESTION ANSWERING

CommonsenseQA (Talmor et al.) is a 5-way multiple choice Question Answering task that requires reasoning with commonsense knowledge, containing 12,102 questions. It can be solved with an pure language model. However, following (Yasunaga et al., 2021). for each problem, we sample a subgraph containing entities in the problem from a knowledge graph. Our baselines include combination of Knowledge graph and LLM: , QAGNN (Yasunaga et al., 2021) and some pure LLM results from (Xu et al., 2021). The results are shown in Table 6. Note that our GL-Fusion is base on Llama3-8b, and our model outperforms Llama3-8b with prompt significantly, verifying that learning on KG indeed significantly boost model's performance.

Table 5: Inductive link prediction on FB15k237-ind dataset. Higher scores are better. KG-BERT serves as a baseline LLM, while others are GNNs. KG-BERT performance is from Zha et al. (2021); GNN baselines are from Zhu et al. (2024).

| Method | v1 | | | v2 | | | v3 | | | v4 | | |
|---|---|---|---|---|---|---|---|---|---|---|---|---|
| | MRR | H@1 | H@10 | MRR | H@1 | H@10 | MRR | H@1 | H@10 | MRR | H@1 | H@10 |
| GraiL | 0.443 | 0.329 | 0.642 | 0.614 | 0.495 | 0.818 | 0.642 | 0.531 | 0.828 | 0.646 | 0.529 | 0.893 |
| NBFNet | 0.625 | 0.519 | 0.834 | 0.769 | 0.673 | 0.949 | 0.762 | 0.668 | 0.951 | 0.774 | 0.681 | 0.960 |
| UniLP | 0.754 | 0.672 | 0.921 | 0.808 | 0.734 | 0.943 | 0.793 | 0.720 | 0.945 | 0.832 | 0.760 | 0.960 |
| KG-BERT | 0.500 | 0.341 | - | - | - | - | - | - | - | - | - | - |
| BertRL | 0.605 | 0.541 | - | - | - | - | - | - | - | - | - | - |
| KRST | 0.716 | 0.602 | - | - | - | - | - | - | - | - | - | - |
| GL-Fusion | **0.8558** | **0.7306** | **0.9830** | **0.8558** | **0.7761** | **0.9853** | **0.8257** | **0.7379** | **0.9861** | **0.8514** | **0.7697** | **0.9905** |

Table 6: Result in accuracy (%) in CommonSenseQA tasks.

| Model | GL-Fusion | QA-GNN | GPT-3 Finetuned | Llama3-8b- |
|---|---|---|---|---|
| ACC | 81.79 | 76.1 | 73.0 | 59.4 |

Table 7: Function name generation task on ogbg-code2 dataset. Baseline results are from the OGB (Hu et al., 2020) leaderboard.

| Model | GAT | GraphTrans | SAT++ | DAGformer | GL-Fusion |
|---|---|---|---|---|---|
| Test F1(%) | $15.69_{\pm0.10}$ | $18.30_{\pm0.24}$ | $22.22_{\pm0.32}$ | $20.18_{\pm0.21}$ | 40.97 |

Table 8: Ablation study on ogbn-arxiv dataset. All results are text accuracy ($\uparrow$)

| | Original | Low Rank | w/o cross atten | w/o gate | w/o aggrs | w/o gnn pred | w/o text pred |
|---|---|---|---|---|---|---|---|
| GNN | **77.09** | 76.61 | 75.5 | 75.88 | 75.4 | - | 75.8 |
| LLM | **76.43** | 75.97 | 74.97 | 73.33 | 75.57 | 72.45 | - |
| Ensemble | **78.2** | 77.06 | 76.35 | 76.2 | 76.48 | - | - |

## 5.5 GRAPH-TO-LANGUAGE GENERATION

While image-to-text and video-to-text generation are well-studied, graph-to-language generation remains underexplored. We evaluate our GL-Fusion model on the ogbg-code2 dataset (Hu et al., 2020), where each graph represents the Abstract Syntax Tree (AST) of a Python function, and the goal is to predict the function name from the AST. Previous methods have treated this as a classification task, assuming uniform function name lengths and a limited token set, which is impractical. In contrast, our model generates text directly. The results are shown in Table 6. Our GL-Fusion model outperforms state-of-the-art methods (Velickovic et al., 2018; Luo et al., 2023) signficantly, verifying the strong potential of our architecture.

## 5.6 ABLATION STUDY

To verify the effectiveness of our designations, we conduct ablation study on ogbn-arxiv dataset. The results are shown in Table 8. In the table, Low rank denotes using model with LoRA rank=8. w/o cross atten removes the graph-text cross attention. w/o gate removes the gate mechanism. w/o aggrs removes multiple aggregators in the message passing modules in transformer layers and uses mean aggregator only. w/o gnn pred removes the gnn prediction and training loss on it. w/o text pred removes the text prediction and training loss on it. Each module contributes to the overall performance, and removing any of them results in a performance drop.

## 6 CONCLUSION

This paper addresses key challenges in integrating GNNs and LLMs, such as independent encoding of text and structure, task-agnostic text representation, excessive information compression, and issues with output scalability and flexibility. We propose the GL-Fusion architecture, featuring three innovations: 1. Integrated Structural and Textual Learning by combining MPNNs with self-attention layers; 2. Graph-text cross-attention modules that preserve full textual content to prevent information loss; and 3. A GNN-LLM twin-predictor that facilitates predictions as both LLM and GNN for enhanced scalability and flexibility. Our experiments on the various datasets demonstrate that GL-Fusion effectively leverages language and structure, significantly improving reasoning performance in graph-text tasks.

## LLM USAGE

We use the LLM to polish our writing and check our grammar over the full paper.

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

# A EXPERIMENTAL DETAILS

**Architecture** All our models are based on Llama-3-8b, which utilizes a 32-layer Transformer, with message passing at layers 0, 4, 8, 12, 16, 20, 24, and 28, and node cross-attention at layers 3, 7, 11, 15, 19, 23, 27, and 31. For the original LLaMA-3 layers, we employ LoRA with rank $r$ and weight $\alpha = 2r$, while newly added layers are trained with full parameters. The LLM used for sentence encoding is LLM2Vec (BehnamGhader et al., 2024) model based on Llama-3-8b, is fine-tuned with Lora on its last layer. For ogbl-arxiv dataset, we use $r = 64$. For CSTAG datasets, we use $r = 4$. For all other datasets, we use $r = 32$. GNN in our model in implemented with torch geometric (Fey & Lenssen, 2019). Some parameters are loaded from pretrained LLM.

- For structure-aware Transformer Layers, we leverage the pretrained Llama-3-8B model as the backbone. As detailed in Section 3.1, we introduce new parameters through modifications to positional encoding and attention masks. Additionally, for the `<graph_node>` token, we simply add an embedding to the existing layer. The newly added MPNN component is initialized randomly. We fine-tune the transformer parameters using the Low-Rank Adaptation (LoRA) method from the peft library. This approach freezes the original transformer layers and optimizes only the newly introduced low-rank layers. The MPNN parameters, on the other hand, are trained entirely from scratch.
- For the Graph-Text Cross-Attention layers in our paper, all parameters are initialized randomly and trained from scratch.
- For the GNN prediction modules, which are simple linear layers for generating outputs, parameters are initialized randomly and trained from scratch.
- Overall, most parameters are initialized from pretrained models and frozen. Only about 10% of the parameters in the entire model are optimized.

**Training Process** All experiments are trained on A800 GPUs with one epoch. The optimizer is AdamW with learning rate=3e-5, weight decay=0.1. For node classification and csqa datasets, we ensemble the prediction of GNN and LLM. For link datasets, we use GNN output only. For graph level tasks and synthetic tasks, we use text output only. The entire model is trained end-to-end, with joint training of all components.

**Baselines** Baselines and experiments setting used in different works on combining GNN and LLM varies a lot. Therefore, we directly use the results reported by baseline works as shown in the maintext.

## B   DATASET DETAILS

### B.1   DETAILS OF FB15K-237-IND DATASETS

Table 9: Statistics of FB15k-237-ind benchmark. (Teru et al., 2020)

|    |          | #links | #nodes | #links |
|----|----------|--------|--------|--------|
| v1 | train    | 183    | 2000   | 5226   |
|    | ind-test | 146    | 1500   | 2404   |
| v2 | train    | 203    | 3000   | 12085  |
|    | ind-test | 176    | 2000   | 5092   |
| v3 | train    | 218    | 4000   | 22397  |
|    | ind-test | 187    | 3000   | 9137   |
| v4 | train    | 222    | 5000   | 33916  |
|    | ind-test | 204    | 3500   | 14554  |

The statistics of the FB15k-237-ind is in Table 9. To generate the text attributes of the datasets, we first map Freebase objects to wikidata[1], and then used the detailed texts from the wikiKGv2-90m dataset in OGB-LSC (Hu et al., 2021) as the textual representations for each node. Like other KG completion work on GNN, we also added reverse relation for each relation and label them as "[reverse] xxx".

The total graph is too large to input the GNN, so like previous work we sample a subgraph. For training set, we first sample random 50 negative samples by perturb the head or tails of a triple and then sample the 3-hop neighbors of these nodes. Then we reduce the subgraph size with limiting the maximal number of nodes is 500.

An example of our dataset:

---

The input question:

---

The Adventures of Tintin (2011 film directed by Steven Spielberg)
–/film/film/production_companies⟶ ?. The graph: `<graph_start><node><node>...<node><graph_end>`.

---

The node text features in the graph:

---

{"title": "Jane Krakowski", "desc": "American actress", "dist to head": 6}
{"title": "Shochiku", "desc": "Japanese movie studio and production company for kabuki.", "dist to head": 3}
{"title": "Erin Brockovich", "desc": "2000 biographical movie by Steven Soderbergh", "dist to head": 2}
. . .

---

The relation text features:

---

{"title": "/film/film/runtime./film/film_cut/film_release_region"}
{"title": "[reverse] /music/record_label/artist"}
. . .

---

### B.2   DETAILS OF CSTAG DATASETS

CSTAG datasets, including children, history, computers. photo, sport, are provided by Yan et al. (2023). We conduct node classification tasks on them. Their statistics are shown in Table 10. We directly use the node text provided by the dataset.

### B.3   DETAILS OF CITATION GRAPHS

We use citation graphs ogbn-arxiv (Hu et al., 2020) and cora (Liu et al., 2023) and classify nodes on them. Their statistics are shown in Table 11. For ogbn-arxiv, we directly use official split. For node text, we use paper title and abstract. We also add label of non-target nodes in training set to input node text. The edge text are "cite" or "cited".U

---

[1] There are some tools to convert FB to wikidata, like `https://github.com/happen2me/freebase-wikidata-convert?tab=readme-ov-file`

Table 10: Statistics of CSTAG datasets.

| Dataset | #Nodes | #Edges | #Class | Split | Ratio | Metric | text length |
|---------|--------|--------|--------|-------|-------|--------|-------------|
| Children | 76,875 | 1,554,578 | 24 | Random | 60/20/20 | Acc | 256 |
| History | 41,551 | 358,574 | 12 | Random | 60/20/20 | Acc | 256 |
| Computers | 87,229 | 721,081 | 10 | Time | 72/17/11 | Acc | 256 |
| Photo | 48,362 | 500,928 | 12 | Time | 60/20/20 | Acc | 512 |
| Sports | 173,055 | 1,773,500 | 13 | Random | 20/10/70 | F1 | 64 |

Table 11: Statistics of citation datasets.

| Dataset | #Nodes | #Edges | #Class | Split | Ratio | Metric | text length |
|---------|--------|--------|--------|-------|-------|--------|-------------|
| ogbn-arxiv | 169,343 | 1,166,243 | 40 | Time | 54/18/28 | Acc | 256 |
| Cora | 2,807 | 5,429 | 7 | Random | 10/10/80 | Acc | 256 |

B.4 DETAILS OF COMMON SENSE QUESION ANSWERING DATASET.

Common Sense Quesion Answering dataset (Talmor et al.) is a dataset answering common sense questions based on knowledge graph. Each question is a 5-way multiple choice task that requires reasoning with commonsense knowledge. The dataset contains 12,102 questions. The split is fixed. The test set of CommonsenseQA is not publicly available, and model predictions can only be evaluated once every two weeks. So we report results on the in-house (IH) data splits used in Yasunaga et al. (2021).

B.5 DETAILS OF GRAPH PROPERTY PREDICTION DATASET.

We use ogbg-code2 (Hu et al., 2020), a dataset containing 158,100 code graphs with each graph containing 125.2 nodes and 124.2 edges on average. The task is to predict the function name of the code represented by the graph. We use node type in compiler and node attribute (values of constants) as the node feature.

