# OpenReview forum: "GL-Fusion: Rethinking the Combination of Graph Neural Network and Large Language model"
_ICLR.cc/2026/Conference — Submitted to ICLR 2026_

### Official Review · Reviewer_vf2z · 2025-10-24

**Soundness:** 3
**Presentation:** 2
**Contribution:** 2
**Rating:** 4
**Confidence:** 3

**Summary:**

This paper presents an architecture called GL-Fusion, which seeks to combine the strengths of Graph Neural Networks and Large Language Models. The authors propose several innovations to address the limitations observed in existing approaches: (1) Structure-Aware Transformers that integrate GNN's message-passing capabilities into LLM's transformer layers, (2) Graph-Text Cross-Attention to ensure that textual information from graph nodes and edges is preserved, and (3) a GNN-LLM Twin Predictor to simultaneously generate outputs from both the GNN and LLM components.

**Strengths:**

1. The proposed GL-Fusion architecture is well thought out and offers improvements over previous methods. The integration of both GNNs and LLMs is innovative, addressing key challenges such as the loss of information during compression in other models.
2. The structure of the paper is logical and easy to follow, with well-explained figures that illustrate the architecture and workflow of GL-Fusion.

**Weaknesses:**

1. The paper's primary weakness is its unclear positioning relative to prior work on GNN-LLM fusion. The "Structure-Aware Transformer" (Section 3.1) seems conceptually very similar to models like GraphFormers (Yang et al., 2023). The authors state that GraphFormers "combines GNNs and Transformers by iteratively encoding text and aggregating graph structures across layers," which sounds functionally similar to the proposed approach. However, GraphFormers is notably absent as a baseline in the experimental comparison (e.g., in Table 2 or Table 7).
2. While the paper demonstrates strong results on several datasets, a broader comparison with additional state-of-the-art methods would be beneficial to further validate the proposed approach. Additionally, there are significant blank spaces in Tables 5 and 8, and Table 7 has formatting issues that need to be addressed.

**Questions:**

See Weaknesses.

---

### Official Review · Reviewer_TVQB · 2025-10-25

**Soundness:** 3
**Presentation:** 2
**Contribution:** 3
**Rating:** 6
**Confidence:** 3

**Summary:**

The paper proposes GL-Fusion, a novel architecture for text-rich graph learning that integrates GNNs and LLMs through three key innovations:

**Structure-Aware Transformers**, which embed GNN-style message passing directly into LLM Transformer layers;

**Graph-Text Cross-Attention**, which preserves full raw text from both nodes and edges during attention computation;

**GNN-LLM Twin Predictor**, a dual-head output module that enables parallel numerical predictionand autoregressive language generation.

**Strengths:**

**Principled Integration of Structural and Textual Reasoning**

The Structure-Aware Transformer elegantly reconciles seemingly conflicting requirements: causal masking for language generation, permutation invariance for graph nodes, and multi-hop message passing for structural awareness. This design directly addresses the core limitations of prior paradigms—LLM-centric models that ignore graph topology and GNN-centric models that lose textual nuance through premature embedding compression.

**Strong Few-Shot Performance:**

GL-Fusion demonstrates significant gains in low-data regimes, particularly on benchmarks like OGBN-Arxiv under 100-shot settings. This suggests the architecture effectively leverages LLM priors while grounding them in graph structure, enabling robust generalization with minimal supervision.

**Broad Task Coverage:**

The framework achieves competitive results across diverse graph tasks—including node classification and link prediction—indicating its flexibility and general applicability to standard graph representation learning problems.

**Weaknesses:**

**Architectural Fragmentation in the Twin Predictor:**

While the GNN-LLM Twin Predictor aims to support heterogeneous output formats, it introduces a modular split that contradicts the current trend toward unified, end-to-end generative frameworks. This dual-head design may hinder the model’s ability to develop a coherent internal representation that jointly reasons about structure and semantics, potentially limiting its capacity for deep cross-modal understanding—a key goal in the LLM-for-graphs community.

**Lack of Context Window Management:**

The paper does not address the practical bottleneck of LLM context length. When scaling to large graphs, the tokenized graph (including full node/edge texts) can easily exceed typical context windows, leading to information truncation or neighborhood subsampling that undermines structural integrity. GL-Fusion appears to assume manageable input sizes, offering no mechanism to handle real-world large-scale graphs.

**Questions:**

see Weakness

And, recent LLM-for-graph works  have expanded beyond node/link tasks to language-intensive graph applications such as commonsense reasoning over knowledge graphs, graph-to-text generation, or interactive graph-based QA. Does GL-Fusion exhibit advantages in these semantically rich, interactive tasks?

---

### Official Review · Reviewer_PagW · 2025-11-01

**Soundness:** 1
**Presentation:** 1
**Contribution:** 1
**Rating:** 0
**Confidence:** 5

**Summary:**

This paper proposes GL-Fusion, an architecture that attempts to integrate Graph Neural Networks (GNNs) and Large Language Models (LLMs) via three components: (1) Structure-Aware Transformers that embed GNN message-passing inside transformer layers, (2) Graph-Text Cross-Attention for connecting textual and structural tokens, and (3) a Twin-Predictor combining GNN and LLM heads for different output formats. The authors claim that GL-Fusion achieves state-of-the-art results on several benchmarks (e.g., ogbn-arxiv, ogbg-code2).

**Strengths:**

1. The motivation of the paper, to more tightly integrate graph and textual representations, is timely and relevant to recent GNN-LLM fusion research.

**Weaknesses:**

1. The paper is difficult to follow structurally. The Related Work section appears after the main methodology (Section 4) rather than preceding it. The introduction contains only one citation, indicating a lack of contextual grounding. Moreover, figure and table layouts are confusing, e.g., Tables 5–8 are visually overlapping and inconsistently formatted, and labels such as “v1/v2/v3/v4” in Table 5 are unexplained anywhere in the text
2. The claimed novelty of Structure-Aware Transformers and Graph-Text Cross-Attention is overstated. The architecture essentially uses a standard GNN to compute node embeddings, then concatenates them with LLM embeddings before prediction. This is not fundamentally different from previous hybrid models such as GraphFormers (Yang et al., 2023) or GraphGPT (Tang et al., 2024).
3. The experimental section is cluttered and lacks rigor. The datasets (e.g., CSTAG, FB15k-237-ind) are not representative for evaluating multimodal reasoning because they include small-scale or outdated tasks. The authors also reuse reported numbers from other papers rather than reimplement baselines, as they admit in Appendix A (“we directly use the results reported by baseline works”)
4. Many of the claims in the paper are inaccurate or lack sufficient evidence. For example, in lines 278–289, the authors discuss the limitations of using LLMs for graph-related tasks, but several of these assertions are questionable. If LLMs were inherently unable to perform numerical reasoning, how do we explain their success in solving mathematical problems where outputs are numeric? Similarly, the statement that LLMs cannot generate predictions in parallel is misleading. While autoregressive models generate tokens sequentially, LLMs can process batches of node representations and produce predictions for multiple nodes within a single forward pass. These examples reflect a pattern of unsupported or misleading claims throughout the manuscript. The authors should provide proper citations or empirical evidence to substantiate each major statement, especially when asserting limitations of LLMs.

**Questions:**

1. What differentiates the “Structure-Aware Transformer” from simply appending message-passing outputs to the token embeddings, as in prior GraphFormers or GraphGPT?
2. How were v1/v2/v3/v4 variants in Table 5 defined—do they represent dataset splits, model versions, or parameter settings?
3. Since you “directly use baseline results from prior papers” (Appendix A), how do you ensure fair experimental comparison under the same evaluation setup?
4. Why are key baselines (e.g., recent sequence-serialization methods or graph-instruction tuned LLMs) missing from comparisons?
5. The reported accuracy improvements are small; have you conducted any statistical significance tests or sensitivity analyses?

---

### Official Review · Reviewer_D6uL · 2025-11-02

**Soundness:** 2
**Presentation:** 1
**Contribution:** 2
**Rating:** 2
**Confidence:** 3

**Summary:**

This paper studies graph learning with LLMs, and proposed GL-Fusion, a new model architecture that integrates message passing into transformer layers. The authors categorized existing works into GNN-centered and LLM-centered ones, and discussed limitations of each. The proposed GL-Fusion contains couple interestingly designed new components: structure-aware Transformer, graph-text cross-attention, GNN-LLM twin predictor. Evaluated showed sota performances on the datasets.

**Strengths:**

1. The proposed design is quite novel to my knowledge. It interestingly proposed to actually integrate message passing into the transformer layers, trying to take the benefit of both.
2. The authors conducted fairly comprehensive evaluation across multiple tasks and benchmarks, with the proposed method outperforming baselines.

**Weaknesses:**

1. While the new model layer design would take the benefit from both message passing layers and Transformer layers, it seems would also inherit scalability limitations from both. The combination of a LLM with message-passing layers is computationally heavy. The paper did not discuss runtime or memory trade-offs.
2. In addition to the above point. The authors mentioned that "Each node is represented by a \<node\> token". This can be very costly and hence non-practical in real applications, where the number of nodes can easily go beyond hundreds of millions.
3. The overall presentation of this paper should be improved. The spacings of sections and tables are not consistent and page 9 even has overlapped text.

**Questions:**

n/a

---

### Meta-Review · Area_Chair_RESB · 2026-01-06

**Summary:**

While the "Structure-Aware Transformer" was recognized for its innovation in balancing language generation with graph structural reasoning, the decision to reject was driven by the following critical flaws highlighted by reviewers:

Architectural Concerns: Reviewers noted that the GNN-LLM dual-head predictor introduces modular fragmentation, contradicting the current trend towards unified end-to-end generation. This design hinders the construction of coherent internal representations, thereby limiting the model's capability for deep cross-modal understanding of both structure and semantics.

Scalability and Efficiency: The integration of LLMs with message-passing layers is computationally expensive. The paper failed to address trade-offs regarding runtime and memory usage, and the model appears to inherit the scalability limitations of both GNNs and Transformers.

Writing Quality: The writing quality was flagged as needing improvement to meet publication standards.

**Reviewer Concerns:**

Addressed Issues: None.
Unresolved Issues: All raised issues remain unresolved.

**Reviewer Scores:**

0.2.4.6

---

### Decision · Program_Chairs · 2026-01-26

Reject